# Diet Quality and Mortality among Chinese Adults: Findings from the China Health and Nutrition Survey

**DOI:** 10.3390/nu16010094

**Published:** 2023-12-27

**Authors:** Jiali Zheng, Tianren Zhu, Fangyu Li, Han Wu, Shuo Jiang, Nitin Shivappa, James R. Hébert, Xiaoguang Li, Yan Li, Hui Wang

**Affiliations:** 1School of Public Health, Shanghai Jiao Tong University School of Medicine, Shanghai 200025, China; marcus_zhu@sjtu.edu.cn (T.Z.); hooleejnn@126.com (S.J.); lixg@shsmu.edu.cn (X.L.); yanli2022@sjtu.edu.cn (Y.L.); 2Department of Epidemiology, Human Genetics and Environmental Sciences, School of Public Health, The University of Texas Health Science Center at Houston, Houston, TX 77030, USA; fli4@tulane.edu; 3Division of Noncommunicable Diseases and Injury, Shanghai Municipal Center for Disease Control and Prevention, Shanghai 200336, China; wuhan@scdc.sh.cn; 4Cancer Prevention and Control Program, Arnold School of Public Health, University of South Carolina, Columbia, SC 29208, USA; shivappa@mailbox.sc.edu (N.S.); jhebert@mailbox.sc.edu (J.R.H.); 5Department of Epidemiology and Biostatistics, Arnold School of Public Health, University of South Carolina, Columbia, SC 29208, USA

**Keywords:** dietary pattern, diet quality, Diet Quality Index-International, Chinese Healthy Eating Index, Dietary Inflammatory Index, mortality, cohort studies, China

## Abstract

The association between diet quality and all-cause mortality in Chinese population is unclear. We aimed to study the associations of three a priori diet quality indices—including the Diet Quality Index-International (DQI-I), Chinese Healthy Eating Index (CHEI), and energy-adjusted Dietary Inflammatory Index (E-DII)—and their included components with all-cause mortality. We used baseline data from the 2004, 2006, 2009, and 2011 waves of the China Health and Nutrition Survey (CHNS). We used a multivariable-adjusted Cox model to examine the associations between DQI-I, CHEI, and E-DII with all-cause mortality. During a mean of 7 years of follow-up, a total of 461 deaths occurred among 12,914 participants. For DQI-I, there were significant inverse associations with mortality for the variety score (HR_Q4 vs_. _Q1_ = 0.69, 95%CI = 0.52–0.92) and overall balance score (HR_>0 vs_. _0_ = 0.81, 95%CI = 0.66–0.91). The adequacy score of CHEI was associated with 40% less risk of all-cause mortality (HR_Q4 vs_. _Q1_ = 0.60, 95%CI = 0.43–0.84). E-DII was not associated with mortality. An estimated 20.1%, 13.9%, and 31.3% of total mortality would be averted if the DQI-I variety score, DQI-I overall balance score, and CHEI adequacy score improved from the bottom to the top quartile, respectively. Improving diet quality, especially improving diet variety and adequacy, and having a more balanced diet may reduce all-cause mortality in Chinese adults.

## 1. Introduction

Diet plays a critical role in shaping human health and well-being. According to the Global Burden of Diseases, Injuries, and Risk Factors Study (GBD) 2017, suboptimal diet is a known modifiable risk factor substantially contributing to non-communicable disease (NCD) mortality, with 11 million deaths across the world attributable to dietary factors [1]. The complexity of human diets has driven the development and application of dietary pattern research that takes into account synergistic effects of different dietary components and, thus, has advantages in predicting disease risks compared with individual food or nutrient study [2]. Abundant evidence has supported an improved overall diet quality, reflected by better dietary index scores and decreased risk of all-cause mortality, as well as mortality due to cardiovascular disease and cancer [3,4,5,6,7,8,9,10]. However, most studies on dietary patterns and mortality were conducted based on a dietary index designed specifically for foods consumed in the Western populations, such as the Healthy Eating Index (HEI), Mediterranean diet score (MDS), and the Dietary Approaches to Stop Hypertension (DASH) [3,11,12]. 

Given the distinctive features of Chinese foods in the world, a dietary index designed for other populations may potentially bias its association with health outcomes in the Chinese population. There was a paucity of research that explored the association between overall dietary quality and mortality in China, where a dietary index fitted with Chinese foods was used, highlighting a significant and unique opportunity to scrutinize the potential mortality risk of Chinese dietary quality assessed by featured dietary indices [13,14,15,16].

Previously there was limited evidence on the association between dietary quality and all-cause mortality in the Chinese population [13,17,18,19,20,21,22,23,24]. Among these, few studies used Chinese diet-related indices to measure diet quality in China, and these studies primarily focused on a single aspect or the overall quality of a diet rather than different aspects of diet quality [13,17,18,19,20,21,22,23,24]. For instance, the Diet Quality Index-International (DQI-I), developed to offer an international standard for assessing overall diet quality which encompasses diet adequacy, variety, overall balance, and moderation [15], and the Chinese Healthy Eating Index (CHEI) which was developed according to the recommendations of the Dietary Guidelines for Chinese-2016 (DGC-2016) were each identified to be associated with all-cause mortality in Hong Kong and Chinese mainland adults [13,16,22]. In addition, the energy-adjusted Dietary Inflammatory Index (E-DII), as a validated measure of the overall inflammatory potential of diet [25], was extensively studied and reported to be positively associated with cancer mortality [26], cardiovascular disease mortality [27], and all-cause mortality [28]; however, in the Chinese population, the E-DII relationship with mortality has not yet been detected. In order to comprehensively cover the main quality aspects of the Chinese diet, we aimed to use these three a priori dietary indices─the DQI-I, CHEI, and E-DII ─ to evaluate their associations, their component scores, and included food or nutrient intake with the risk of all-cause mortality in the representative national cohort of the China Health and Nutrition Survey (CHNS). We also estimated the proportion of deaths in the population that could be attributed to the lowest Chinese diet quality.

## 2. Materials and Methods

### 2.1. Study Design and Study Population

The China Health and Nutrition Survey is a national longitudinal household-based survey initiated in 1989, and it was designed to examine a series of economic, sociological, demographic, and health questions at the individual, household, and community level [29]. A multistage, random cluster process was used in the CHNS to draw the sample from nine provinces from northeast to southwest: Heilongjiang, Liaoning, Jiangsu, Shandong, Henan, Hubei, Hunan, Guizhou, and Guangxi. Nine waves of the survey data were released which covered 1991, 1993, 1997, 2000, 2004, 2006, 2009, 2011, and 2015. The response rate across these surveys was 79–94% at the household level, and 80–88% at the individual level. Details of the study design were described elsewhere [29,30]. All participants provided written informed consent. The ethics approval and consent were approved by the Institutional Review Committees of the University of North Carolina at Chapel Hill, the National Institute for Nutrition and Health, and the Chinese Center for Disease Control and Prevention (approval code: No. 201524). 

We selected Chinese Food Composition Table (FCT) 2002 and 2004, which were combined in this study to cover most nutrients calculated in the dietary indices; therefore, baseline dietary data from the survey waves 2004, 2006, 2009, and 2011 were used. After including 18,589 Chinese residents aged over 18 with available dietary records from 2004 to 2011, we excluded participants if they (1) did not have follow-up information (*n* = 3898); (2) reported missing cooking oil intake (*n* = 1474); (3) did not report person-days on household condiment intake (*n* = 187); and (4) reported an implausible daily energy intake or an implausible body mass index (BMI)—defined as 2 interquartile ranges below the sex-specific 25th percentile or above the 75th percentile of the Box–Cox transformed values (*n* = 116). The final sample size in this study was 12,914 with 7405, 1148, 1531, and 2830 individuals enrolled at year 2004, 2006, 2009, and 2011, respectively. Figure 1 demonstrates the inclusion and exclusion procedures of selecting the study population.

### 2.2. Assessment of Dietary Intake

Participants’ dietary intake data were collected by trained nutritionists on 3 consecutive days (2 weekdays and 1 weekend day) at both household and individual levels. At the individual level, participants completed 3-day 24-hour dietary recalls by reporting all foods and beverages consumed at home and away from home on the previous day. Household food consumption on the same 3 days as the dietary recalls was estimated by calculating the food inventory changes using a weighing and measuring method [31]. Condiment consumption (including salt, cooking oil, soy sauce, and monosodium glutamate) measurements from the household food inventory and individual dietary intake based on dietary recalls were added together to calculate the total dietary intake for each participant, and the average consumption over the three days represented an individual’s daily intake in the present study [32,33]. We linked the daily dietary intake data to FCT 2002 and 2004 to calculate the daily intake of total energy and nutrients. 

### 2.3. Calculation of Scores of Three Dietary Indices

We applied the DQI-I, CHEI, and E-DII to the CHNS in order to evaluate the overall diet quality of Chinese adults from various aspects, including variety, adequacy, moderation, overall balance, and inflammatory potential.

The DQI-I is a validated dietary index in the Chinese population that captures variability in intake of food and nutrients for global comparisons and for exploring aspects of diet quality related to the nutrition transition [15]. DQI-I reflects diet quality from 4 major components: variety, adequacy, moderation, and overall balance [15]. To compute the DQI-I score, we converted food intake in grams to serving sizes with reference to the USDA pyramid servings database and Dietary Guidelines for Americans 2005 [34,35]. Details of the scoring algorithm of DQI-I are described in Appendix A. Briefly, the variety component included the overall variety of five food groups (meat/poultry/fish/eggs; dairy/beans; grain; fruits; vegetables) and the within-group variety of protein sources (meat, poultry, fish, eggs, dairy, beans), with a score ranging from 0 to 20 points. For the adequacy assessment, eight healthy dietary factors (vegetable group, fruit group, grain group, fiber, protein, iron, calcium, vitamin C) were included with each scored from 0 to 5. We modified the three highest energy intake levels corresponding to three maximum serving sizes for the vegetable group, fruit group, and grain group, and the fiber intake as 0–2200 kcal, 2200–2700 kcal, ≥2700 kcal, because one third of our study population consumed a daily total energy of below 1700 kcal, which was the original lowest energy intake level. Total fat, saturated fat, cholesterol, sodium, and empty calorie foods constituted the moderation component with a score of 0 to 30 points. The fourth component was overall balance (macronutrient ratio and fatty acid ratio), with a score ranging from 0 to 10 points. The total DQI-I score ranged from 0 to 100, and higher scores indicated a better diet quality [36].

The DII was designed to quantify the inflammatory potential of an individual’s diet [25]. Details of the development of the DII were described elsewhere [25]. Briefly, eligible research articles published until 2010 were searched to derive the inflammatory effect scores of 45 dietary factors included in the DII (i.e., DII components) [25]. The energy-adjusted daily dietary intake calculated from the CHNS was first standardized to a worldwide dietary database, and the energy-adjusted standardized amount was then multiplied by the inflammatory effect score for each DII component and then summed across all components to obtain the overall DII score for each participant [25]. Energy-adjusted DII (E-DII) scores are computed similarly but are based on intakes adjusted to 1000 kcal and use an energy-adjusted global database. DII and E-DII scores are scored similarly. So, results across studies are comparable. Usually, the choice to use the one is based on model goodness of fit for total model explanatory ability. In general, the E-DII fit the data better about two thirds of the time. Higher E-DII scores represent more pro-inflammatory diets while lower (i.e., more negative) E-DII scores indicate more anti-inflammatory diets. The DII score has been found to be significantly positively associated with IL-6, TNF-α receptor 2, and high-sensitivity CRP in various populations [37,38]. In this longitudinal analysis, 31 out of 45 food parameters, except caffeine, eugenol, folic acid, saffron, turmeric, flavan-3-ol, flavones, flavonols, flavonones, anthocyanidins, isoflavones, oregano, rosemary, and vitamin D, were available and used to calculate the E-DII score. 

The CHEI was specifically designed to measure dietary adherence among the Chinese population to the updated Dietary Guidelines for Chinese (DGC-2016) and has demonstrated good (content and construct) validity and internal consistency in the CHNS study population [16,22,39]. Details of the development and scoring method were provided elsewhere [16]. Briefly, the CHEI was used to assess diet quality from two perspectives: adequacy and limitation, with 12 and 5 food groups included in each dimension, respectively. The individual dietary intake of 13 food groups (12 adequacy foods and 1 limitation food) derived from the CHNS was firstly converted to standard portions (SP) as per the DGC-2016, with one SP of each food group sharing similar content of energy and macronutrient profiles (Appendix A). Compared with the latest version of HEI (HEI-2015), which corresponds to the 2015–2020 Dietary Guidelines for Americans, CHEI emphasized red meat, cooking oils, and alcohol in the limitation section (HEI-2015 emphasized saturated fat and refined grains) and emphasized poultry, eggs, and dark vegetables in the adequacy section, making it uniquely relevant to a traditional Chinese food context [40]. All of the 17 food components in the CHEI were scored from 0 to 5, except for 0 to 10 for fruits, cooking oils, and sodium, and the scores reflecting the amount of intake between the minimum and maximum cutoffs were prorated linearly. The total CHEI score ranged from 0 to 100 with a higher score reflecting better adherence to DGC-2016 (Appendix A). 

### 2.4. Ascertainment of Deaths

Participants’ mortality status and the date of death were reported on the household questionnaires at each wave [41]. If someone was dead, the family member was asked for the date of death. When a case of death was repeatedly reported, the initial death date was counted in the analysis. Specific death causes were not reported in the CHNS during 2004 to 2015. 

### 2.5. Assessment of Covariates

Self-reported information on age, sex, educational level, marital status, household income per year (inflated to 2015), self-reported health status, smoking and alcohol drinking status, physical activity, and personal medical history was collected at all waves of surveys. Weight and height were measured by trained technicians using standard methods [22]. The Body Mass Index (BMI) was calculated as weight (kg)/height (m)^2^ and was divided into underweight (<18.5 kg/m^2^); normal (18.5–23.9 kg/m^2^); overweight (24.0–27.9 kg/m^2^); and obesity (≥28 kg/m^2^) according to the recommended criteria for the Chinese population [42]. Educational level was divided into three categories: primary school and lower; junior and senior middle school; and college and higher. Household income was categorized to low, medium, and high levels based on the tertiles of the distribution. Smoking status was categorized as never, former, and current smokers, while alcohol drinking status was dichotomized as never or ever. Participants self-reported their frequency and duration spent on various types of activities (i.e., domestic, occupational, transportation, and leisure activity), and the total metabolic equivalent (MET) in hours per week was calculated by multiplying the average hours spent on each activity per week by the MET for that activity based on the Compendium of Physical Activities and then categorized into tertiles [43]. Self-reported health status was categorized into four groups (i.e., good and above, fair, poor and below, unknown) based on participants’ responses to the question regarding their perceived health status compared with others of the same age. A history of hypertension, diabetes, cancer, myocardial infarction, and stroke was defined for each by self-reported physician diagnosis or treatments received (for diabetes and hypertension only). A composite measure of the history of comorbidities including these diseases was used in the analysis and was coded as “yes” if an individual had any positive medical history of these diseases.

### 2.6. Statistical Analysis

Baseline characteristics were described and compared between those in the lowest and highest quartiles for the DQI-I, CHEI, and E-DII scores, using means and standard errors for continuous variables and number and frequencies for categorical variables. To examine the correlations between the three diet indices as well as the component scores, Pearson correlation coefficients between each of two scores were calculated, with *p*-values reported.

Person-years of follow-up for each participant were calculated from the baseline (the completion date of the baseline wave questionnaire) to the date of death, lost to follow-up, or to the end of the study in 2015, whichever came first. Cox proportional hazards regression, with person-years as the time metric, was used to analyze the associations between the total scores and component scores of the three dietary indices and all-cause mortality. Hazard ratios (HRs) and 95% confidence intervals (CIs) were estimated with participants in the lowest quartile of each dietary index score as the referent, except for the overall balance component of the DQI-I where HRs and 95% CIs were calculated comparing participants with scores above 0 to those with 0 points, as the majority scored 0 for this component. Two Cox models were applied in the analysis: The crude model was adjusted for age and sex, and the multivariable model was additionally adjusted for predefined confounders including educational level, marital status, baseline year, household income level, total energy, smoking status, alcohol drinking status, BMI status, physical activity level, and a history of comorbidities based on the previous literature [3,21,22]. Because the CHEI and E-DII included alcohol as a component, drinking status was not adjusted in the multivariable model for these two dietary indices. The linear trend of all-cause mortality across quartiles of each dietary index score or its component score was tested using a respective continuous variable for the score, after confirming that the linear assumption was sufficient based on the restricted cubic spline test [44]. The HR and 95% CI for each one-unit increase in each dietary quality score associated with total mortality were also analyzed. The proportional hazard assumption was examined using the Schoenfeld residual test, and no violation was detected in any Cox models [45]. To understand the role that each index component within CHEI and DQI-I may play in the total score and mortality association, we examined the multivariable-adjusted association between the score for each index component and all-cause mortality. 

We further estimated the proportion of deaths that could be attributed to the lowest diet quality as assessed with the lowest quartile of the total score and component scores of the DQI-I and CHEI and the highest quartile of the E-DII score by using the partial population attributable fraction (PAF) while controlling for all potential confounders in the multivariable model. The bootstrap method with 1000 resamples was used to compute the 95% CIs [46,47]. 

A priori stratified analyses of the associations between the continuous scores of the dietary indices and all-cause mortality by sociodemographic characteristics, lifestyle factors, and a history of comorbidities were conducted in the multivariable model. We also conducted the stratified analysis by the baseline year (2004 and 2006–2011) to detect period effects of food availability and the population’s dietary habits over those years. The *p*-value for multiplicative interaction was calculated by adding the cross-product of each dietary score and effect modifier in the multivariable Cox model. Forest plots were used to display results.

We conducted a series of sensitivity analyses. Firstly, we eliminated participants who were dead within the first three years of follow-up to reduce impact by lethal diseases which might substantially influence individuals’ dietary habits. Secondly, a complete case analysis was conducted to eliminate effects of missing data. Thirdly, we assessed the associations between the dietary indices and all-cause mortality in the multivariable Cox model without adjustment of the BMI status. Fourthly, we removed participants with histories of comorbidity at the baseline. Fifthly, we excluded participants with a daily energy intake of less than 1700 kcal since this measurement was the minimum energy level needed for scoring the maximum adequacy scores of the vegetable group, fruit group, and grain group, and the fiber intake in the DQI-I calculation. Additionally, the nutrient profile for calculating one SP varied by food groups in the CHEI. Thus, we uniformly used total energy as the reference (i.e., Energy per SP) to calculate the number of SPs for all food groups in the study. 

All analyses were performed using SAS software (version 9.4, Cary, NC, USA). All tests were two-sided, with *p*-values < 0.05 considered to be statistically significant.

## 3. Results

The baseline score ranges of the DQI-I, CHEI, and E-DII were 24.2 to 82.8, 17.2 to 88.5, and –4.3 to 4.2, respectively. The average daily individual intake of food groups, including dietary fiber, vegetables, fruits, dairy, soybeans, fish and seafood, poultry, and seeds and nuts, was higher in the highest quartile of the DQI-I and CHEI scores but lower with an increasing E-DII quartile (i.e., more pro-inflammatory), while grains and red meat were higher with an increasing quartile of the DQI-I and E-DII. Participants enrolled in 2004 compared with later years were more likely to have higher diet quality as assessed with the DQI-I but lower quality as assessed with the CHEI and E-DII. Participants with higher DQI-I and CHEI scores or lower E-DII scores were more likely to be normal or overweight, more physically active, have a higher educational level, a higher household income, and to have drunk alcohol when compared with participants with the lowest diet quality (Table 1). Correlations between the dietary indices revealed a significant positive link between the DQI-I and CHEI, and the adequacy scores within these two indices (correlation coefficient = 0.45) and the limitation/moderation scores (correlation coefficient = 0.49) were both positively correlated with each other (*p* < 0.01). The E-DII scores displayed a significantly negative correlation with the DQI-I and CHEI scores, particularly with the adequacy scores of these two indices (*p* < 0.01). As for correlations within the dietary indices, all four DQI-I component scores were significantly positively correlated with the DQI-I total score, with coefficients ranging from 0.34 to 0.74. Similarly, both the adequacy and limitation scores of the CHEI were strongly correlated with the total score (Appendix A). 

### 3.1. Associations between Dietary Indices and All-Cause Mortality

During a mean of seven years of follow-up, a total of 461 deaths occurred. After adjustment of all the covariates, no association was identified between the DQI-I total score and all-cause mortality. When looking into the DQI-I components, a significant lower risk of death was associated with a higher level of diet variety (HR_Q4 vs_. _Q1_ = 0.69, 95%CI = 0.52–0.92) and overall balance (HR_>0 vs_. _0_ = 0.81, 95%CI = 0.66–0.99) with a significant 4% and 7% lower all-cause mortality associated with each one-score increment, respectively. On the contrary, a higher score in moderation significantly increased all-cause mortality (HR_Q4 vs_. _Q1_ = 1.35, 95%CI = 1.03–1.77). No association was found between the DQI-I adequacy score and all-cause mortality. In the multivariable-adjusted association between the CHEI and all-cause mortality, no association was observed with the total CHEI score, but the highest quartile of the adequacy score was associated with 40% less all-cause mortality risk (HR_Q4 vs_. _Q1_ = 0.60, 95%CI = 0.43–0.84), while a higher limitation score indicating less intake of foods in this dimension significantly increased all-cause mortality (*P*-trend = 0.03). No association was observed between the E-DII score and all-cause mortality (Table 2).

### 3.2. Associations between Component Scores of Dietary Indices and All-Cause Mortality 

Upon examining the specific components of the dietary indices, we found that in the DQI-I variety dimension, the increased overall food group variety and within-group variety of protein sources were significantly related to 4% (95%CI = 1–8%) and 8% (95%CI = 2–13%) decreased all-cause mortality, respectively. With regard to the moderation component, a higher score, indicating lower cholesterol intake, was associated with higher all-cause mortality (HR_continuous_ = 1.05, 95%CI = 1.01–1.10). A diet balanced in the macronutrient ratio but not in the fatty acid ratio significantly reduced death risk (*p* = 0.04). An increased intake of fruits (HR_continuous_ = 0.93, 95%CI = 0.88–0.98) and eggs (HR_continuous_ = 0.93, 95%CI = 0.88–0.97) under the adequacy score of the CHEI significantly reduced all-cause mortality (Figure 2).

### 3.3. Population Attributable Fraction (PAF) of All-Cause Mortality Owing to the Lowest Diet Quality

There would be an estimated 20.1% (95%CI = 9.3–30.5%) and 13.9% (95%CI = 1.8–25.7%) reduction of total mortality if the DQI-I variety score and the DQI-I overall balance score increased from the lowest to the highest quartile, respectively. If individuals’ diet quality as assessed with the CHEI adequacy score improved from the lowest to the highest quartile, a significant 31.3% (95%CI = 14.6–46.3%) of total death reduction would occur (Table 3).

### 3.4. Stratified Associations between Dietary Indices and All-Cause Mortality 

In the stratified analyses of the continuous associations between the three dietary scores and all-cause mortality, no significant link was identified between the DQI-I score and mortality in all subgroups (Appendix A). In terms of the CHEI, we observed significant effect modifications by BMI status (*P*-interaction = 0.02), baseline year (*P*-interaction = 0.01), and physical activity level (*P*-interaction = 0.02). The significant inverse associations between a higher CHEI and lower mortality were observed only among participants who started follow-up later than 2004 (HR = 0.96, 95%CI = 0.93–0.99), were overweight or obese (HR = 0.97, 95%CI = 0.95–0.99), and/or had the lowest level of physical activity (HR = 0.98, 95%CI = 0.96–0.99) (Appendix A). We detected a significant positive association of the E-DII and mortality among overweight and obese participants (HR = 1.19, 95%CI = 1.01–1.41) but an inverse association in the normal weight group (HR = 0.88, 95%CI = 0.78–0.98) (*P*-interaction = 0.01). In addition, the association was also modified by the baseline year (*P*-interaction = 0.04) (Appendix A).

### 3.5. Sensitivity Analysis

In all of the sensitivity analyses, results did not change materially. In the complete case analysis and the analysis of excluding baseline comorbidities, the significant positive associations between the DQI-I moderation scores, CHEI limitation scores, and all-cause mortality disappeared (Appendix A). Significant associations with the DQI-I and CHEI component scores were generally held when all SPs were calculated with reference to the energy standard, or when participants were eliminated with a daily energy intake of less than 1700 kcal to completely meet the DQI-I scoring criteria, except the non-significant association between the DQI-I moderation score and all-cause mortality (Appendix A).

## 4. Discussion

In this large prospective Chinese cohort, the total scores of three a priori dietary indices showed no significant associations with all-cause mortality, but key components including a greater diet variety, a more balanced diet, better food adequacy, and a higher consumption of fruits and eggs were associated with reduced all-cause mortality. A significant 20.1%, 13.9%, and 31.3% reduction of total mortality would occur in the Chinese population if the DQI-I variety score, DQI-I overall balance score, and CHEI adequacy score improved from the bottom to the top quartile, respectively. 

The non-significant associations between all-cause mortality and the total scores of the three dietary indices we observed could be attributed to the following reasons: in terms of the construction of the DQI-I and CHEI, the sum of the different internal component scores constituted the total score. Therefore, the opposite directions between some of the internal sections in the association with all-cause mortality, for instance, between the variety and moderation section in the DQI-I, and between the adequacy and limitation section in the CHEI, might neutralize the direction of association between the total dietary score and mortality. Secondly, the three dietary indices each took various food components into consideration, making each index a complex integrity. In the whole dietary pattern, how food or nutrients biologically interact with each other to exert a combined effect on mortality remained unclear. Even so, we still discovered an inherent consistency in the associations between the overall and component scores with mortality among the three indices. For instance, our data suggested that an increase in egg consumption (an adequacy food in the CHEI) could reduce the risk of death. This was consistent with our findings that higher cholesterol intake as represented by a lower cholesterol score (a moderation component in the DQI-I) was associated with a lower mortality risk, as dietary cholesterol comes in a large proportion from eggs in the diets of Chinese adults [48]. Within the E-DII framework, cholesterol is recognized for its potent pro-inflammatory potential. Thus, our CHEI findings of the non-significant protective effect of a high-inflammatory diet on mortality were also consistent with the above-mentioned inverse association between cholesterol/eggs and all-cause mortality. However, inconsistency also existed. For instance, adequacy measured from the CHEI significantly reduced death risk, but adequacy from the DQI-I did not, mainly because the CHEI was developed based on food, which has a greater mortality risk compared with the nutrients on which the DQI-I was based.

A similar association between diet quality and all-cause mortality was identified in several previous studies. Qu et al. applied the dietary diversity score (DDS) in 6737 subjects participating in the CHNS study between 2004 and 2015 and identified a 22% reduced all-cause mortality per one-point increase in the DDS [49]. In another CHNS study with a median follow-up of 9.0 years, the variety score of protein sources, defined as the number of proteins consumed at the appropriate level and accounting for both the type and quantity of proteins, had a significant inverse association with overall mortality risk (per score increment, HR = 0.69; 95%CI = 0.66–0.72) [50]. Both of these findings were consistent with our results for an inverse association between the DQI-I variety score and mortality. Food variety substantially influences an individual’s overall nutritional status, and a lack of food variety may lead to malnutrition and a higher mortality rate [51]. Protein intake with adequate variety from various food sources can improve the abundance and diversity of amino acid composition and can thus optimize physiological and metabolic functions, as well as reduce aging-related frailty. These processes are all conducive to a decreased risk of death [52,53]. An overall balanced diet as assessed by ratios between three macronutrients and between three types of fatty acids (poly-unsaturated fatty acids (PUFA), mono-unsaturated fatty acids (MUFA) and saturated fatty acids) was observed in our study to be protective against mortality risk. Consistently, diets with high carbohydrate levels, low fat, and low protein were reported to be associated with an increased lifespan and improved cardiometabolic outcomes in late life [54]. In the Nurses’ Health Study and the Health Professionals Follow-up Study, replacing 5% of energy from saturated fats with equivalent energy from PUFA and MUFA was associated with an estimated 27% and 13% reduced total mortality risk, respectively [55]. However, excessive PUFAs are susceptible to lipid peroxidation, leading to excessive oxidative stress, inflammation, and cancer, thereby highlighting the need for a balanced and appropriate proportion of unsaturated fat in the diet [56]. 

We observed significant associations between adequacy measured from the CHEI and reduced death risk, and, based on its component analyses, a greater intake of fruits and eggs was found to be the main driver. The inverse association between fruit intake and total death risk was consistently demonstrated in the literature [57,58,59,60]; in a systematic review of 64 meta-analyses, the highest identified linear dose response for each 100 g/day increase in fruit intake was 0.87 (95%CI = 0.84–0.90) for all-cause mortality [57]. A high consumption of fruits which contain a wide range of antioxidant compounds and polyphenols—such as Vitamin C, carotenoids, and flavonoids—has been shown to prevent the oxidation of cholesterol and other lipids in the arteries, inhibit platelet aggregation, and reduce vascular tone to prevent CVD, the largest contributor to all-cause mortality [60,61]. The association between egg consumption and mortality was, however, debatable. Our data suggested a decreased total mortality risk with increasing egg consumption, which was supported by similar findings in another CHNS study of 18,914 adults, and dietary cholesterol from eggs in this study was also found to be inversely associated with total mortality (*P*-trend = 0.001), consistent with our findings of the cholesterol score in the DQI-I [62]. A nonlinear inverse association of egg intake and total mortality among 134,280 Chinese adults in the Shanghai Women’s Health Study and the Shanghai Men’s Health Study also confirmed our findings [63]. But, a recent meta-analysis which comprised 33 cohort studies with a mixture of worldwide populations found that egg intake was not associated with the risk of mortality from all causes (RR _the highest versus lowest level_ = 1.02, 95%CI = 0.94–1.11) [64]. A meta-analysis concluding nine studies on dietary cholesterol intake and mortality demonstrated a significantly increased mortality risk with cholesterol intake in the US studies but no associations in the non-US studies [65]. These discrepant egg/cholesterol–mortality associations across populations may be due to differences in dietary patterns and lifespan disease status across sociodemographics and ethnicities [64]. More evidence is warranted for the effects of dietary cholesterol from different foods on health outcomes, especially those from long-term randomized controlled trials and large prospective cohort studies among Chinese general populations. Mechanistically, many beneficial components in eggs could account for the protective effect of egg intake on mortality, such as high-quality protein, omega-3 poly-unsaturated fatty acids, and lecithin.

Our findings indicated a remarkable reduction in mortality with improvements in diet quality, as measured by both the DQI-I and CHEI. A shift from the lowest to the highest quartile of the DQI-I variety and balance score would result in an estimated 20.1% and 13.9% total mortality reduction, respectively. More strikingly, having the best adherence to the highest quartile of the CHEI adequacy score would eliminate 31.3% of total death risk in China. These findings emphasized the capability of dietary quality improvements related to diet variety, balance, and adequacy to reduce death risk, independent of other modifiable factors. Jessri et al. used PAF to estimate the mortality burden of poor dietary patterns as measured by several Western dietary scores, including the HEI-2015, DASH, and MDS, and reported that 26.5–38.9% (men) and 8.9–22.9% (women) of deaths were attributable to poor dietary patterns in Canada [66]. According to the GBD study, age-standardized proportions of all-cause mortality attributable to dietary risks among Chinese adults were 30.2%; a high intake of sodium was the leading dietary risk for deaths in 2017 in China [1], but no study so far has reported the PAF of deaths attributable to dietary patterns or quality aspects in China. Our findings of the great reductions in total mortality are particularly important when considering the size of the Chinese population and the potential number of lives that could be saved with dietary improvements, signifying that enhancing diet quality holds substantial promise for overall survival in China. 

We provided first-line evidence in a representative Chinese prospective cohort to examine how Chinese diet quality was associated with all-cause mortality by utilizing three dietary indices encapsulating multiple diet aspects. Using detailed dietary measurements from three consecutive-day 24-h food recalls and household inventories to calculate three indices based on Chinese DRIs, DGC-2016 recommendations, and 31 nutrient-based dietary factors provided a valid and relatively accurate representation of Chinese dietary patterns. Our approach extends beyond analyzing the total diet quality score alone. By analyzing each component score or component food/nutrient within the DQI-I and CHEI, we have gleaned a more profound understanding of the impact of crucial elements in affecting mortality risk. The observed internal and inter-index consistency in the multivariable-adjusted mortality associations, along with the unchanged results from sensitivity analyses, underscores the robustness of our findings. Limitations also existed. The first limitation presented itself in diet-related measurement errors. The three-day food recall and household food inventory had inevitable recall-related or measurement errors, and diet information was only collected in one week and only once at baseline, which may have compromised the long-term diet representativeness. Fourteen food parameters were not available in the E-DII calculation, and all of these components were anti-inflammatory, which could have led to a potential underestimation of the associations between the E-DII and mortality. For the outcome ascertainment, self-reported death information collected in the survey may have involved measurement errors as well. All of these measurement errors related to exposure and outcome were likely to be non-differential and, therefore, are likely have biased the results toward null. Another limitation was residual or unmeasured confounding. In addition, due to data limitation, we cannot estimate the relationship between dietary patterns and cause-specific mortality. The absence of incident disease diagnoses and important serum biomarkers further complicates the interpretation of the underlying mechanism of diet patterns on mortality. Hence, future studies with such information and refined diet intake measurements are required to replicate and enrich the mortality findings and yield possible mechanisms.

## 5. Conclusions

No association was identified between all-cause mortality and the total DQI-I, CHEI, or E-DII scores. Better dietary quality, including a better variety of food, an overall balanced diet, and better food adequacy, can reduce the risk of all-cause mortality among Chinese adults. A 20.1%, 13.9%, and 31.3% of mortality in China could be prevented through improving food variety, overall balance, and adequacy, respectively, to an optimal level, highlighting the necessity for enhancing public health efforts to improve diet quality at the population level in China. Future studies based on Chinese dietary indices and refined diet measurements are warranted to explore the associations with all-cause and cause-specific mortality in the Chinese population. 

## Figures and Tables

**Figure 1 nutrients-16-00094-f001:**
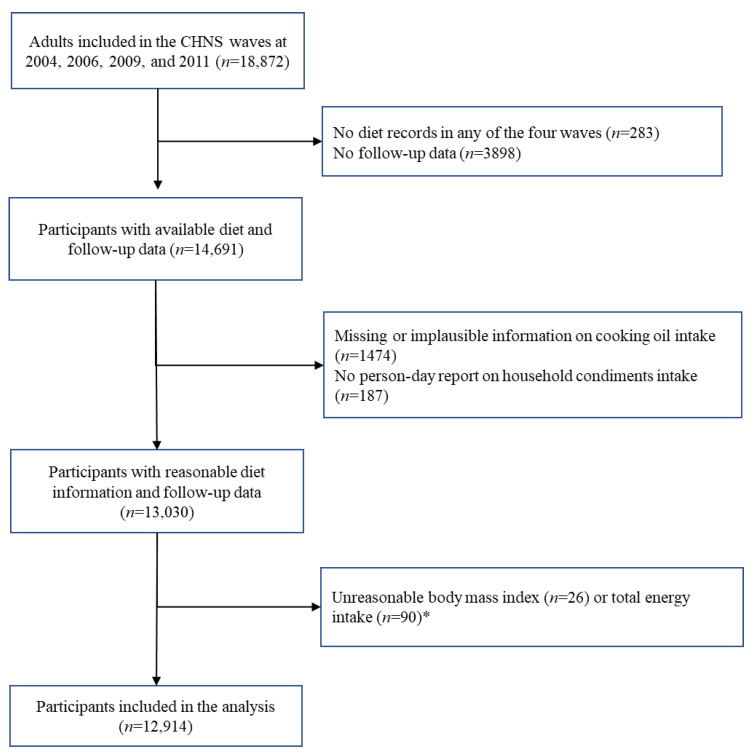
Flow chart of study population selection. * Outliers were defined as 2 interquartile ranges below the sex-specific 25th percentile or above the 75th percentile of the Box–Cox transformed respective variables.

**Figure 2 nutrients-16-00094-f002:**
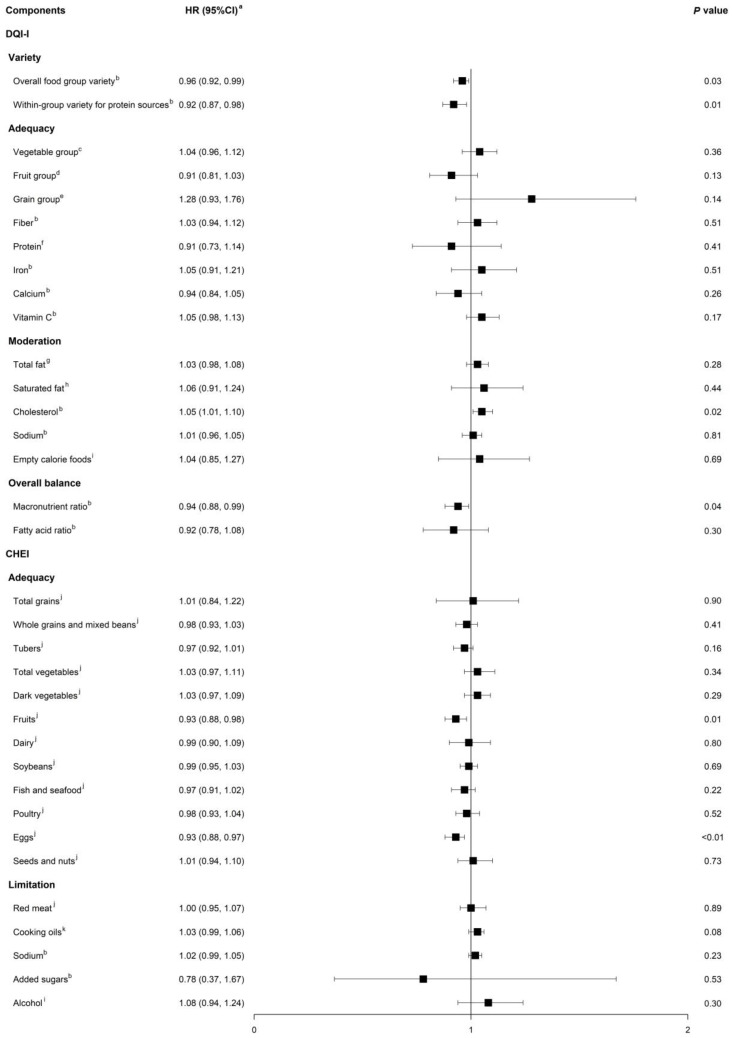
Multivariable-adjusted associations between scores of specific components in the DQI-I and CHEI with all-cause mortality in the CHNS. Abbreviations: CHEI, Chinese Healthy Eating Index; DQI-I, Dietary Quality Index-International. ^a^ The hazard ratios indicated the change in the risk of death associated with a 1-unit increase in each score of components in the CHEI and DQI-I. ^b^ Associations of mortality with overall food group variety, within-group variety for protein sources, fiber, iron, calcium, Vitamin C, cholesterol, sodium, macronutrient ratio, and the fatty acid ratio were computed in Model 2 as described in Table 1. ^c^ Scores of the fruit group, grain group from the DQI-I, and dairy, soybeans, fish and seafood, poultry, eggs, seeds and nuts from the CHEI were additionally adjusted on the basis of Model 2. ^d^ Scores of the vegetable group, grain group from the DQI-I, and dairy, soybeans, fish and seafood, poultry, eggs, seeds and nuts, and red meat from the CHEI were additionally adjusted on the basis of Model 2. ^e^ Scores of the fruit group, vegetable group from the DQI-I, and dairy, soybeans, fish and seafood, poultry, eggs, seeds and nuts, and red meat from the CHEI were additionally adjusted on the basis of Model 2. ^f^ Score of total fat from the DQI-I and carbohydrate intake was additionally adjusted on the basis of Model 2. ^g^ Score of protein from the DQI-I and carbohydrate intake was additionally adjusted on the basis of Model 2. ^h^ Score of protein from the DQI-I and carbohydrate, mono-unsaturated fat, and poly-unsaturated fat intake was additionally adjusted on the basis of Model 2. ^i^ Scores of protein, total fat from the DQI-I, and carbohydrate intake were additionally adjusted on the basis of Model 2. ^j^ Scores of all other food groups from the CHEI were mutually adjusted on the basis of Model 2. ^k^ Scores of total grains, tubers, total vegetables, fruits, dairy, soybeans, fish and seafood, poultry, eggs, seeds and nuts, and red meat from the CHEI were additionally adjusted on the basis of Model 2.

**Table 1 nutrients-16-00094-t001:** Baseline characteristics of 12,914 participants by quartiles of DQI-I, CHEI, and E-DII in the CHNS study.

Characteristics	DQI-I		CHEI		E-DII	
	Q1	Q4	Q1	Q4	Q1	Q4
** *n* **	3229	3229	3229	3228	3228	3229
	**Mean (SE)**	**Mean (SE)**	**Mean (SE)**	**Mean (SE)**	**Mean (SE)**	**Mean (SE)**
**Age (years)**	48.72 (0.28)	45.17 (0.25)	48.56 (0.27)	46.09 (0.27)	48.38 (0.27)	46.04 (0.27)
**Dietary factors**						
Total energy (kcal/d)	1841.94 (14.88)	2296.04 (12.71)	2096.02 (15.84)	2048.55 (11.84)	1934.42 (13.10)	2228.86 (13.57)
Carbohydrate (g/d)	234.32 (1.71)	357.88 (2.36)	282.05 (2.12)	295.75 (2.15)	273.33 (2.10)	316.52 (2.34)
Protein (g/d)	51.83 (0.40)	75.70 (0.53)	54.46 (0.38)	73.66 (0.45)	68.26 (0.51)	59.71 (0.38)
Fat (g/d)	77.96 (1.21)	65.17 (0.50)	82.47 (1.24)	68.34 (0.50)	63.83 (0.72)	82.29 (0.83)
Dietary fiber (g/d)	7.37 (0.08)	14.31 (0.18)	8.86 (0.13)	12.72 (0.14)	14.06 (0.20)	8.03 (0.07)
Grain group (g/d)	321.92 (2.46)	440.61 (3.49)	382.22 (2.95)	367.08 (3.07)	352.21 (2.90)	415.62 (3.26)
Vegetable group (g/d)	31.42 (0.40)	44.88 (0.66)	38.54 (0.50)	39.00 (0.54)	44.65 (0.71)	34.15 (0.41)
Fruit group (g/d)	8.84 (0.37)	29.68 (0.74)	5.47 (0.32)	35.96 (0.66)	24.83 (0.66)	10.13 (0.41)
Dairy group (g/d)	10.08 (0.52)	14.06 (0.55)	3.45 (0.27)	24.52 (0.65)	14.85 (0.55)	7.51 (0.42)
Soybean group (g/d)	15.84 (0.45)	28.34 (0.55)	14.02 (0.43)	29.93 (0.54)	26.98 (0.60)	16.41 (0.44)
Fish and seafood group (g/d)	12.19 (0.41)	23.12 (0.54)	9.96 (0.38)	27.00 (0.56)	21.50 (0.53)	12.82 (0.41)
Red meat group (g/d)	17.60 (0.33)	20.68 (0.34)	19.45 (0.36)	19.61 (0.31)	18.42 (0.32)	20.97 (0.35)
Poultry group (g/d)	8.20 (0.37)	11.83 (0.42)	3.69 (0.28)	17.49 (0.45)	11.66 (0.40)	7.90 (0.35)
Egg group (g/d)	14.91 (0.28)	12.72 (0.25)	10.02 (0.24)	18.21 (0.36)	14.12 (0.32)	13.01 (0.25)
Seeds and nuts group (g/d)	2.52 (0.21)	2.93 (0.16)	0.75 (0.09)	5.90 (0.24)	4.32 (0.22)	0.95 (0.09)
	***n* (%)**	***n* (%)**	***n* (%)**	***n* (%)**	***n* (%)**	***n* (%)**
**Baseline year**						
2004	1720 (53.27)	1801 (55.78)	2169 (67.17)	1256 (38.91)	1592 (49.32)	2018 (62.50)
2006	239 (7.40)	290 (8.98)	265 (8.21)	249 (7.43)	233 (7.22)	298 (9.23)
2009	313 (9.69)	484 (14.99)	261 (8.08)	506 (15.68)	379 (11.74)	376 (11.64)
2011	957 (29.64)	654 (20.25)	534 (16.54)	1226 (37.98)	1024 (31.72)	537 (16.63)
**Sex**						
Male	1899 (58.81)	1537 (47.60)	1810 (56.05)	1713 (53.07)	1785 (55.30)	1672 (51.78)
Female	1330 (41.19)	1692 (52.40)	1419 (43.95)	1515 (46.93)	1443 (44.70)	1557 (48.22)
**Educational level**						
Primary school and lower	1257 (38.93)	1001 (31.00)	1462 (45.28)	691 (21.41)	1084 (33.58)	1217 (37.69)
Junior and senior middle school	1707 (52.74)	1890 (58.53)	1614 (49.98)	2004 (62.08)	1777 (55.05)	1801 (55.78)
College and higher	258 (7.99)	324 (10.03)	135 (4.18)	523 (16.20)	347 (10.75)	195 (6.04)
Missing	11 (0.34)	14 (0.43)	18 (0.56)	10 (0.31)	20 (0.62)	16 (0.50)
**Marital status**						
Married	2619 (81.11)	2763 (85.57)	2636 (81.64)	2742 (84.94)	2681 (83.05)	2660 (82.38)
Not in a marriage ^a^	590 (18.27)	444 (13.75)	568 (17.59)	466 (14.44)	517 (16.02)	546 (16.91)
Missing	20 (0.62)	22 (0.68)	25 (0.77)	20 (0.62)	30 (0.93)	23 (0.71)
**Household income per year ^b^**						
Low, CNY < 16,962	1070 (33.14)	901 (27.90)	1358 (42.06)	574 (17.78)	915 (28.35)	901 (27.90)
Medium, CNY 16,962–39,590	1124 (34.81)	1096 (33.94)	1136 (35.18)	979 (30.33)	993 (30.76)	1096 (33.94)
High, CNY ≥39,590	1002 (31.03)	1211 (37.50)	685 (21.21)	1658 (51.36)	1291 (39.99)	1211 (37.50)
Missing	33 (1.02)	21 (0.65)	50 (1.55)	17 (0.53)	29 (0.90)	34 (1.05)
**Self-reported health status**						
Good and above	1137 (35.21)	1310 (40.57)	1377 (42.64)	950 (29.43)	1066 (33.02)	1415 (43.82)
Fair	655 (20.28)	658 (20.38)	840 (26.01)	473 (14.65)	624 (19.33)	737 (22.82)
Poor and below	157 (4.76)	106 (3.28)	198 (6.13)	69 (2.14)	119 (3.69)	144 (4.46)
Missing	1280 (39.64)	1155 (35.77)	814 (25.21)	1736 (53.78)	1419 (43.96)	933 (28.89)
**Smoking status**						
Never	2307 (71.45)	2132 (66.03)	2214 (68.57)	2293 (71.03)	2242 (69.45)	2198 (68.07)
Former	108 (3.34)	131 (4.06)	94 (2.91)	134 (4.15)	138 (4.28)	103 (3.19)
Current	802 (24.84)	951 (29.45)	909 (28.15)	792 (24.54)	833 (25.81)	909 (28.15)
Missing	12 (0.37)	15 (0.46)	12 (0.37)	9 (0.28)	15 (0.46)	19 (0.59)
**Drinking status**						
Never	2242 (69.43)	1995 (61.78)	2192 (67.88)	2091 (64.78)	2079 (64.41)	2171 (67.23)
Ever	974 (30.16)	1223 (37.88)	1027 (31.81)	1131 (35.04)	1139 (35.29)	1039 (32.18)
Missing	13 (0.40)	11 (0.34)	10 (0.31)	6 (0.19)	10 (0.31)	19 (0.59)
**BMI status**						
Underweight (<18.5 kg/m^2^)	164 (5.08)	168 (5.20)	186 (5.76)	151 (4.68)	159 (4.93)	167 (5.17)
Normal (18.5–24.0 kg/m^2^)	1580 (48.93)	1688 (52.28)	1652 (51.16)	1592 (49.32)	1661 (51.46)	1685 (52.18)
Overweight (24.0–28.0 kg/m^2^)	963 (29.82)	918 (28.43)	873 (27.04)	1043 (32.31)	954 (29.55)	887 (27.47)
Obese (≥28.0 kg/m^2^)	342 (10.59)	266 (8.24)	305 (9.45)	323 (10.01)	296 (9.17)	291 (9.01)
Missing	180 (5.57)	189 (5.85)	213 (6.60)	119 (3.69)	158 (4.89)	199 (6.16)
**Physical activity level ^c^**						
Low, <52.6 (MET h/wk)	1134 (35.12)	841 (26.05)	1130 (35.00)	835 (25.87)	951 (29.46)	994 (30.78)
Medium, 52.6–146.6 (MET h/wk)	918 (21.43)	1079 (33.42)	762 (23.60)	1319 (40.86)	1136 (35.19)	879 (27.22)
High, ≥146.6 (MET h/wk)	801 (24.81)	1046 (32.39)	947 (29.33)	857 (26.55)	848 (26.27)	999 (30.94)
Missing	376 (11.64)	263 (8.14)	390 (12.08)	217 (6.72)	293 (9.08)	357 (11.06)
**History of comorbidities ^d^**						
No	2709 (83.90)	2820 (87.33)	2809 (86.99)	2684 (83.15)	2655 (82.25)	2850 (88.26)
Yes	489 (15.14)	381 (11.80)	378 (11.71)	517 (16.02)	533 (16.51)	343 (10.62)
Missing	31 (0.96)	28 (0.87)	42 (1.30)	27 (0.84)	40 (1.24)	36 (1.11)

Abbreviations: BMI, Body Mass Index; CHEI, Chinese Healthy Eating Index; DQI-I, Dietary Quality Index-International; E-DII, energy-adjusted Dietary Inflammatory Index; SE, Standard Error. ^a^ The “Not in a marriage” status included never married, divorced, widowed, or separated. ^b^ The household income at baseline was conceptualized as the sum of all sources of income and revenue minus expenditures, which was finally inflated to 2015 CNY/Yuan currency values. ^c^ Total metabolic equivalent (MET) in hours per week was calculated by multiplying the average hours spent on each activity per week by MET for that activity based on the Compendium of Physical Activities and then categorized into tertiles. ^d^ History of comorbidities including history of hypertension, diabetes, myocardial infarction, and stroke. Participants were considered as having hypertension if they reported a history of diagnosis or a history of using antihypertensive drugs. Participants were considered as having diabetes if they reported a history of diagnosis or a history of receiving any of the treatments (special diet, weight control, oral medicine, insulin injection, Chinese traditional medicine, home remedies, qi gong). Participants were considered as having myocardial infarction or stroke if they reported any history of diagnosis.

**Table 2 nutrients-16-00094-t002:** Associations between DQI-I, CHEI, and DII scores and their components and all-cause mortality in 12,914 participants in the CHNS study.

	Quartile 1	Quartile 2	Quartile 3	Quartile 4	*P-_trend_* ^a^	HR_continuous_ (95%CI) ^b^
**DQI-I**						
Total score, mean (range)	44.01 (24.18–49.09)	51.72 (49.10–54.06)	56.42 (54.07–58.87)	63.06 (58.88–82.82)		
Events/person-years	116/21,111	131/22,781	130/23,744	84/22,966		
Model 1, HR (95%CI) ^c^	Ref.	1.27 (0.98, 1.63)	1.25 (0.97, 1.62)	0.94 (0.70, 1.24)	0.32	0.99 (0.98, 1.01)
Model 2, HR (95%CI) ^d^	Ref.	1.29 (1.00, 1.68)	1.31 (1.01, 1.70)	1.03 (0.77, 1.39)	0.99	1.00 (0.99, 1.01)
**Variety**						
Total score, mean (range)	6.89 (3.00–9.00)	10.10 (10.00–11.00)	12.77 (12.00–14.00)	16.68 (15.00–20.00)		
Model 1, HR (95%CI) ^c^	Ref.	1.04 (0.81, 1.34)	0.83 (0.65, 1.07)	0.56 (0.43, 0.74)	<0.01	0.94 (0.92, 0.96)
Model 2, HR (95%CI) ^d^	Ref.	1.03 (0.80, 1.33)	0.89 (0.69, 1.15)	0.69 (0.52, 0.92)	<0.01	0.96 (0.94, 0.99)
**Adequacy**						
Total score, mean (range)	20.99 (9.43–23.93)	25.58 (23.94–27.07)	28.40 (27.08–29.80)	32.14 (29.81–40.00)		
Model 1, HR (95%CI) ^c^	Ref.	0.84 (0.65, 1.07)	0.88 (0.68, 1.13)	0.84 (0.65, 1.09)	0.18	0.99 (0.97, 1.01)
Model 2, HR (95%CI) ^d^	Ref.	0.94 (0.73, 1.21)	0.96 (0.74, 1.25)	1.07 (0.80, 1.43)	0.58	1.01 (0.98, 1.03)
**Moderation**						
Total score, mean (range)	7.37 (0–9.00)	12.00 (12.00–12.00)	15.00 (15.00–15.00)	19.43 (18.00–27.00)		
Model 1, HR (95%CI) ^c^	Ref.	1.02 (0.76, 1.37)	1.22 (0.92, 1.63)	1.70 (1.31, 2.20)	<0.01	1.05 (1.03, 1.07)
Model 2, HR (95%CI) ^d^	Ref.	0.92 (0.68, 1.24)	1.00 (0.75, 1.34)	1.35 (1.03, 1.77)	0.01	1.03 (1.01, 1.05)
**Overall balance ^e^**						
Total score, mean (range)	0	3.11 (2.00–10.00)				
Model 1, HR (95%CI) ^c^	Ref.	0.84 (0.69, 1.03)			0.09	0.95 (0.90, 1.01)
Model 2, HR (95%CI) ^d^	Ref.	0.81 (0.66, 0.99)			0.02	0.93 (0.88, 0.99)
**CHEI**						
Total score, mean (range)	36.82 (17.23–42.12)	45.22 (42.13–48.18)	51.14 (48.19–54.45)	60.97 (54.46–88.51)		
Events/person-years	154/24,122	146/23,881	102/23,186	59/19,144		
Model 1, HR (95%CI) ^c^	Ref.	1.14 (0.91, 1.43)	0.88 (0.68, 1.13)	0.63 (0.47, 0.85)	<0.01	0.98 (0.97, 0.99)
Model 2, HR (95%CI) ^d^	Ref.	1.16 (0.92, 1.45)	0.98 (0.76, 1.26)	0.90 (0.66, 1.23)	0.30	0.99 (0.98, 1.01)
**Adequacy**						
Total score, mean (range)	13.41 (2.80–16.82)	19.06 (16.83–21.23)	23.89 (21.24–26.96)	33.31 (26.97–55.86)		
Model 1, HR (95%CI) ^c^	Ref.	0.94 (0.74, 1.17)	0.75 (0.59, 0.96)	0.40 (0.29, 0.55)	<0.01	0.96 (0.95, 0.97)
Model 2, HR (95%CI) ^d^	Ref.	0.96 (0.76, 1.20)	0.87 (0.67, 1.11)	0.60 (0.43, 0.84)	<0.01	0.98 (0.96, 0.99)
**Limitation**						
Total score, mean (range)	18.91 (7.40–22.81)	24.88 (22.82–26.82)	28.55 (26.83–30.03)	32.13 (30.04–35.00)		
Model 1, HR (95%CI) ^c^	Ref.	1.27 (0.98, 1.64)	1.35 (1.04, 1.75)	1.28 (0.97, 1.67)	0.03	1.02 (1.00, 1.04)
Model 2, HR (95%CI) ^d^	Ref.	1.25 (0.96, 1.62)	1.34 (1.04, 1.75)	1.29 (0.98, 1.70)	0.03	1.02 (1.00, 1.04)
**E-DII**						
Mean (range)	–0.45 (−4.28–0.47)	0.91 (0.48–1.31)	1.66 (1.32–2.02)	2.50 (2.03–4.21)		
Events/person-years	107/21,023	124/22,561	111/23,578	119/23,440		
Model 1, HR (95%CI) ^c^	Ref.	1.17 (0.90, 1.51)	1.04 (0.80, 1.35)	1.06 (0.82, 1.38)	0.52	1.03 (0.95, 1.11)
Model 2, HR (95%CI) ^d^	Ref.	1.03 (0.79, 1.33)	0.89 (0.68, 1.16)	0.85 (0.65, 1.11)	0.21	0.95 (0.88, 1.03)

Abbreviations: CHEI, Chinese Healthy Eating Index; CI, confidence interval; DII, Dietary Inflammatory Index; DQI-I, Dietary Quality Index-International; HR, hazard ratio. ^a^ The *p*-value for trend was obtained from models with indices or component scores as continuous variables in the adjusted Cox models. ^b^ The hazard ratio indicated the change in the risk of death associated with a 1-unit change in each dietary index and component score. ^c^ Model 1 adjusted for age and sex (male, female). ^d^ Model 2 adjusted for age, sex (male, female), baseline year (2004, 2006, 2009, 2011), educational level (primary school and lower, junior and senior middle school, college and higher, missing), marital status (married, not in a marriage, missing), household income (CNY < 13,174, CNY 13,174–48,855, CNY ≥ 48,855, missing), total energy intake, smoking status (never, former, current, missing), drinking status (never, ever, missing), body mass index (<18.5 kg/m^2^, 18.5–24.0 kg/m^2^, 24.0–28.0 kg/m^2^, ≥28.0 kg/m^2^), physical activity level (<52.6 MET h/wk, 52.6–146.6 MET h/wk, ≥146.7 MET h/wk), and a history of comorbidities (no, yes, missing). Since the CHEI and E-DII included alcohol as a component, drinking status was not adjusted in Model 2 for these two scores. ^e^ Since the majority got zero points for the “overall balance” component, the hazard ratio was calculated by comparing participants who scored greater than 0 with those who scored 0.

**Table 3 nutrients-16-00094-t003:** Partial population attributable fractions of all-cause mortality attributed to the lowest diet quality in the CHNS.

Exposures ^a^	Partial PAF (95%CI) ^b^
**DQI-I total score**	**5.3% (−10.9%, 21.2%)**
Variety score	20.1% (9.3%, 30.5%)
Adequacy score	3.7% (−12.7%, 19.9%)
Moderation score	NA ^d^
Overall balance score ^c^	13.9% (1.8%, 25.7%)
**CHEI total score**	**13.0% (−5.4%, 30.6%)**
Adequacy score	31.3% (14.6%, 46.3%)
Limitation score	NA ^d^
**E-DII score**	**NA ^d^**

Abbreviations: CHEI, Chinese Healthy Eating Index; CI, confidence interval; DQI-I, Dietary Quality Index-International; E-DII, energy-adjusted Dietary Inflammatory Index; NA, not applicable; PAF, population attributable fraction. ^a^ The lowest quartiles of total scores and component scores for the DQI-I and CHEI and the highest quartile of the E-DII score were treated as the eliminated exposures as compared with the highest diet quality in the respective PAF calculation, which indicated the reduction proportion of total deaths in the entire population if people’s diet qualities improved from the lowest to the highest level. ^b^ Partial PAF was applied to control other covariates as potential confounders in the multivariable model (Model 2). ^c^ Due to the overall balance score having been dichotomized to above 0 and 0, we calculated the partial PAF of deaths when those with an overall balance score of 0 improved. ^d^ NA, The partial PAF results were not estimable because the adjusted HR_Q4 vs Q1_ for the DQI-I and CHEI scores was above 1 and the adjusted HR_Q4vsQ1_ was below 1 for the E-DII in our study.

## Data Availability

The longitudinal datasets used for analysis are open access and can be obtained at https://www.cpc.unc.edu/projects/china/data/datasets/longitudinal (accessed on 19 August 2021).

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
