# Peer review of "Diet Quality and Mortality among Chinese Adults: Findings from the China Health and Nutrition Survey"

_nutrients, 2023, doi:10.3390/nu16010094_

Round 1

Reviewer 1 Report

Comments and Suggestions for Authors

The present study investigated the relationship between diet quality indices and mortality using the China Health and Nutrition Survey. The authors identified inverse associations of DQI-I variety score, DQI-I overall balance score, and CHEI adequacy score with mortality. However, no associations were found between DQI-I, CHEI, E-DII scores, and mortality. The findings are based on a representative national cohort in China. Nevertheless, several points, as indicated below, need to be addressed by the authors to improve the article's quality.

1.                 The Introduction and Discussion sections lack sufficient description of previous studies on DQI-I, CHEI, and E-DII. Additionally, many diet quality scores already exist, and the strength of DQI-I, CHEI, and E-DII should be indicated.

 2.                 The authors determined participants' mortality status and date of death using self-reported information. The precision of self-reporting should be discussed.

 3.                 Although no associations were observed between DQI-I total, CHEI total, and E-DII scores and all-cause mortality, the authors did not discuss the reasons for these no of associations. Additionally, the conclusion was not based on these findings. Please discuss and conclude the study based on these results.

 4.                 Line 496: Reference 57 examined the total HEI-2015, DASH, and MDS scores. Results from a component of the diet quality index cannot be directly compared to those from total diet quality scores.

 5.                 Abstract: Please provide the number of participants.

Reviewer 2 Report

Comments and Suggestions for Authors

I have a few remarks.

Please provide a detailed description of the purpose of your work.

First of all, the introduction should be more developed.

Do the authors have the consent of the bioethics committee? Please quote this.

The tables are not very legible, too much information in one table (tables 1,2,3,4).

Lack of part -  The Limitations of the study with critical points of view.

The conclusion should more readable and should be develop.

Reviewer 3 Report

Comments and Suggestions for Authors

The manuscript entitled "Diet Quality and Mortality among Chinese Adults: Findings from the China Health and Nutrition Survey" represents original research conducted within the Chinese population. Its primary objective is to investigate the associations between three a priori diet quality indices - DQI-I, CHEI, E-DII - and their respective components with all-cause mortality.

This manuscript holds significance for the scientific community as it provides novel insights into the study of this population group. It also offers valuable information aimed at improving the mortality rate in China through enhancements in the dietary patterns of the population.

The bibliography is up-to-date and of high quality.

However, there are some changes that I mention below, which I believe will enhance the final version of the manuscript:

 - References 1 and 58 relate to the same article. They should be merged and rewritten according to the journal's guidelines.

I suggest:

Afshin, A.; Sur, P.J.; Fay, K.A.; Cornaby, L.; Ferrara, G.; Salama, J.S.; Mullany, E.C.; Abate, K.H.; Abbafati, C.; Abebe, Z., et al. GBD 2017 Diet Collaborators. Health effects of dietary risks in 195 countries, 1990–2017: a systematic analysis for the Global Burden of Disease Study 2017. Lancet 2019, 393, 1958-1972, doi:https://doi.org/10.1016/S0140-6736(19)30041-8

 - The reference 52 should be verified.

I suggest:

Harikrishnan, S.; Jeemon, P.; Mini, G.; Thankappan, K.; Sylaja, P. GBD 2017 Causes of Death Collaborators. Global, regional, and national age-sex-specific mortality for 282 causes of death in 195 countries and territories, 1980-2017: a systematic analysis for the Global Burden of Disease Study 2017. Lancet 2018 Nov 10;392(10159):1736-1788. doi: 10.1016/S0140-6736(18)32203-7. 

- I think DOIs should be added to several references, namely 26, 27, 35, 38, 41, 45, 53, 54, 55, 57, and 59.

I leave it to the authors' consideration.

Round 2

Reviewer 1 Report

Comments and Suggestions for Authors

Thank you for revising your manuscript according to the reviewer's comments. The manuscript was revised well. I don't have any comments on the revised manuscript.